# Agarwood Chromone Alleviates Gastric Ulcers by Inhibiting the NF-κB and Caspase Pathways Based on Network Pharmacology and Molecular Docking

**DOI:** 10.3390/ph18040514

**Published:** 2025-03-31

**Authors:** Canhong Wang, Yulan Wu, Bao Gong, Junyu Mou, Xiaoling Cheng, Ling Zhang, Jianhe Wei

**Affiliations:** 1Hainan Provincial Key Laboratory of Resources Conservation and Development of Southern Medicine, Key Laboratory of State Administration of Traditional Chinese Medicine for Agarwood Sustainable Utilization, Hainan Branch Institute of Medicinal Plant Development, Chinese Academy of Medical Sciences and Peking Union Medical College, Haikou 570311, China; xinzhuangjianpo@163.com (C.W.); wyl18789066212@163.com (Y.W.); gongbao0112@aliyun.com (B.G.); mjy5197664@163.com (J.M.); 18162350450@163.com (X.C.); 2Guangdong Key Laboratory of Green Agricultural Products Processing and Intelligent Equipment, Guangdong University of Petrochemical Technology, Maoming 525099, China; 3School of Pharmacy, Hainan Medical University, Haikou 570300, China; 4National Engineering Laboratory for Breeding of Endangered Medicinal Materials and Key Laboratory of Bioactive Substances and Resources Utilization of Chinese Herbal Medicine, Institute of Medicinal Plant Development, Chinese Academy of Medical Sciences and Peking Union Medical College, Beijing 100193, China

**Keywords:** agarwood chromone, gastric ulcer, anti-inflammation, anti-apoptosis, molecular docking

## Abstract

**Background:** Agarwood has been widely used for the treatment of gastrointestinal diseases. Our research group has suggested that agarwood alcohol extracts (AAEs) provide good gastric mucosal protection. However, the exact mechanisms underlying this effect remain unclear. **Objectives:** This study aimed to investigate the ameliorative effect of agarwood chromone on gastric ulcers and its mechanism. **Methods:** Network pharmacology was used to predict the disease spectrum and key therapeutic targets of 2-(2-phenylethyl)chromone (CHR1) and 2-(2-(4-methoxyphenyI)ethyl)chromone (CHR2). Mice were orally administered CHR1 (20 and 40 mg/kg) and CHR2 (20 and 40 mg/kg) and the positive drug omeprazole as an enteric-coated capsule (OEC, 40 mg/kg) orally. After 7 days of pretreatment with the CHRs, gastric ulcers were induced using absolute ethanol (0.15 mL/10 g). The ulcer index, gastric histopathology, biochemical parameters, and inflammatory and apoptotic proteins were evaluated. Finally, binding of the core compounds to the key targets was verified via molecular docking and visualized. **Results:** The pharmacological results show that the CHRs reduced the gastric occurrence and ulcer inhibition rates by up to more than 70% in a dose-dependent manner. The CHRs decreased the levels of interleukin 6 (IL-6), interleukin 12 (IL-12), interleukin 18 (IL-18), and tumor necrosis factor α (TNF-α), and improved the severity of the pathological lesions in the gastric tissue. The expression of ATP-binding box transporter B1 (ABCB1), arachidonic acid-5-lipoxygenase (ALOX5), nuclear factor kappa B (NF-κB), cysteinyl aspartate specific proteinase 3 3 (Caspase3), and cysteinyl aspartate specific proteinase 9 (Caspase9) was inhibited, but the expression of B-cell lymphoma-2 (Bcl-2) was enhanced. The CHRs bound stably to the key targets via hydrogen bonding, van der Waals forces, etc. These results demonstrate that agarwood chromone compounds exert alleviative effects against the occurrence and development of gastric ulcers by inhibiting the NF-κB and caspase pathways. The CHRs have a therapeutic effect on gastric ulcers through anti-inflammation and anti-apoptosis mechanisms. **Conclusions:** This study suggests that agarwood may have a potential role in drug development and the prevention and treatment of gastrointestinal inflammation, and tumors.

## 1. Introduction

Agarwood, *Aquilaria* spp. (Thymelaeaceae), is a fragrant wood containing resin. It is a famous traditional Chinese medicine (TCM) [1] that has been used for more than one thousand years for the treatment of various diseases, particularly gastrointestinal diseases [2,3]. It has been shown that sesquiterpenes and 2-(2-phenylethyl) chromone derivatives are two of the predominant constituents of agarwood [4,5]. Our previous study showed that the components of agarwood extract include sesquiterpenes (10.615%), chromone (31.678%), aromatics (31.831%), and other known compounds (25.760%), and that chromone is a predominant constituent of agarwood [6]. Agarwood has been reported to have anti-inflammatory and pain relief properties, in addition to other biological activities, and it has been used to treat painful and inflammatory diseases, such as gastric ulcers and gastritis, in clinical settings [7,8]. However, the effect and mechanism of agarwood in the treatment of gastrointestinal diseases remain unclear. Agarwood extracts, essential oils, and primary compounds have exhibited extensive pharmacological properties, including sedative [9,10], anti-neuroinflammatory [11], laxative [12], antioxidant [13,14], anti-inflammatory [15,16], and antibacterial [17] effects. Clinical applications have found that agarwood has a significant curative effect in the treatment of peptic ulcers, stomachache, and functional dyspepsia [18,19,20]. Several studies, including ours, have reported that agarwood extract can relieve intestinal tonic spasms, provide gastric ulcer mucosal protection, and inhibit the occurrence of ethanol-induced gastric ulcers in rats and mice [21,22,23,24]. Chromone is one of the main medicinal components of agarwood extract. Among its derivatives, 2-(2-phenylethyl) chromone and 2-(2-(4-methoxyphenyI)ethyl)chromone are the most abundant compounds and have similar structures [6]. Numerous studies, including ours, have found that chromones and their derivatives have significant anti-inflammatory and gastric cell protective effects [15,16,22,23,24]. It can be inferred that agarwood chromones may protect against gastric ulcers. However, the effect and mechanism of agarwood chromones in preventing the formation of gastric ulcers has not been verified by pharmacological studies. This study compares the effects of these two chromones against gastric ulcers. 

Gastric ulcers are a common and frequent gastrointestinal disease that seriously endangers human health and affects approximately 10% of people; the factors contributing to their pathogenesis are complex and varied [25]. The main risk factors for gastric ulcers include alcohol, smoking, and the use of non-steroidal anti-inflammatory drugs. [26]. Excessive alcohol intake increases mucosal permeability, mucosal damage, cell necrosis, and the inflammatory responses [27]. The mechanisms of gastric injury have not been fully clarified, but it is well known that pro-inflammatory mediators, including IL-6, IL-12, TNF-α, and NF-κB, play an important role in the development of ulcers [28]. Furthermore, cell apoptosis is a major defensive system in the gastric mucosa [29]. At present, synthetic drugs have good efficacy for gastric ulcer therapy. However, their long-term use can cause serious side-effects [30,31]. Therefore, this study aims to find safe and effective natural drugs for the treatment of gastric ulcers.

This study aims to investigate the effect of CHRs on ethanol-induced gastric ulcers, explore their potential mechanisms, and analyze and discuss the protective effect and possible mechanism of CHRs on ethanol-induced gastric mucosal epithelial cell injury. This research provides scientific data and reference for further study of agarwood’s protective mechanisms of action and the development of innovative drugs to treat gastrointestinal inflammatory diseases.

## 2. Results

### 2.1. Target Disease Spectra and Mechanism Signaling Pathways of Agarwood Chromone Compounds

The target spectra of the chromone compounds were predicted, and their target diseases and mechanisms of action were analyzed. As can be seen from the bar chart of the spectral enrichment analysis results (Figure 1A–D), the two chromone compounds have similar therapeutic profiles and mechanisms of action. The chromone compounds may be able to prevent and cure infectious diseases, inflammatory diseases, autophagy, atherosclerotic heart disease, cancer, etc. Their mechanisms of action mainly include interleukin signaling, nuclear receptor transcription signaling, fatty acid metabolism, and biological oxidation. Screening the spectra of target diseases and mechanisms of action showed that agarwood chromone could be used to prevent and treat inflammatory diseases due to its anti-inflammatory, anti-oxidant, and anti-apoptotic effects, providing a reference for this study on the treatment and prevention of gastric ulcers.

### 2.2. Targets of Agarwood Chromone Compounds 

The set of genes related to inflammation and apoptosis were searched and extracted from the Gene Cards database. The results were integrated and the frequency was calculated. The higher the Total Relevance score, the higher the reliability of data derived from multiple search terms. By integrating the target data from the above 10 sources and standardizing them, the potential targets of the two chromone compounds were predicted. There are a total of 60 potential targets for 2-2-phenylethyl (2-(2-phenylethyl)chromone, CID: 441964) (Appendix A) and 63 potential targets for 2-[2-(4-methoxyphenyi)ethyl)chromone, CID: 185208) (Appendix A). Due to the high structural similarity between the two compounds, they have more common targets relating to inflammation and apoptosis (Table 1).

### 2.3. Effect of CHRs on Gastric Lesions and Ulcer Inhibition Rates

Gastric lesions on the gastric mucosal surface were determined by measuring the ulcer index. Figure 2a(B) shows the model group of gastric ulcers induced using ethanol. Compared with the normal group, the gastric tissues of the model mice showed obvious ulcer damage with significantly increased ulcer index (28.69 ± 4.36) in the tissue shown in Figure 2b, which is characterized by gastric mucosal epithelial cell injury, linear hemorrhages, and ulceration craters in the mucosal layer. After the CHRs treatment, damage to gastric epithelial cells, mucosal bleeding and gastric ulcers were relieved in a dose-dependent manner (Figure 2a(C–G)), suggesting that the CHRs exert a protective effect against alcohol-induced gastric cell injury. Fortunately, pretreatment with the CHRs reduced ethanol-induced mucosal damage; the ulcer index in Figure 2b compares the model group (*p* < 0.001). The ulcer inhibition rate showed the protective effect of CHRs against gastric ulcers (Figure 2c). The ulcer inhibition rates were enhanced in a dose-dependent manner for the CHR1 and CHR2 (50.51%, 66.16%, and 59.60%, 74.24%), and omeprazole enteric-coated capsule (OEC) (79.80%) groups. These results showed that the CHRs had a significant inhibitory effect on the occurrence of gastric ulcers, and that the effect of CHR2 is superior to that of CHR1, with a lower ulceration index and higher ulcer inhibition rate compared to CHR1.

### 2.4. Effect of CHRs on Tissue Damage

To observe the degree of damage to the gastric tissue, macroscopic gastric biopsies were performed, as shown in Figure 3. Compared to the normal group, the administration of anhydrous ethanol induced significant gastric tissue damage, including submucosa edema, hemorrhagic injury, mucosal degradation, epithelial cell loss, inflammatory cell infiltration, and necrosis (Figure 3B). Pretreatment with CHRs considerably reduced these changes in the gastric mucosa and provided protection against gastric lesions and injury to gastric mucosal epithelial cells. Mice administered OEC presented a small swelling caused by anhydrous ethanol, with lower numbers of inflammatory cells (Figure 3C). CHR1 and CHR showed a dose-dependent protection on the gastric mucosa (Figure 3D–G). A histogram of the histopathological damage index shows the results more intuitively (Figure 3H).

### 2.5. Effect of CHRs on Inflammatory Cytokines Production

To evaluate the anti-inflammatory effect of the CHRs on ulcers, we measured the levels of pro-inflammatory cytokines, including TNF-α, IL-6, IL-18, and IL-12 in the serum. As shown in Figure 4A–D, the levels of pro-inflammatory cytokines were significantly increased in the mice with gastric ulcers (*p* < 0.05 or *p* < 0.01) compared to the normal group. The CHR treatment dramatically attenuated the anhydrous ethanol-induced elevation of these pro-inflammatory cytokines (*p* < 0.05 or *p* < 0.01). The results showed that the CHRs can inhibit the secretion of pro-inflammatory cytokines, and suggest that agarwood chromone compounds have a good anti-inflammatory effect.

### 2.6. CHRs Inhibit the Expression of ABCB1, ALOX5, NF-κB, Caspase9, and Bcl-2

To further explore the underlying anti-inflammatory and anti-apoptotic molecular mechanisms of the CHRs, we performed IHC to detect the levels of ABCB1, ALOX5, NF-κB, Caspase9, and Bcl-2 (Figure 5a–e) in the stomach tissue of the mice with gastric ulcers. The results showed that the levels of ABCB1, ALOX5, NF-κB, and Caspase9 (Figure 5a–d) were significantly increased in the mice with gastric ulcers in the model group, but pretreatment with the CHRs reduced these expressions in a dose-dependent manner. In contrast, the expression of Bcl-2 was significantly down-regulated in the model group (Figure 5e(B)), while the level of the Bcl-2 was increased dose-dependently by the CHR pretreatments (Figure 5e(C–G)). Quantitative analysis diagrams demonstrate this more clearly (Figure 5e(H)). These results suggest that the CRHs evidently exert anti-inflammatory and anti-apoptotic effects by down-regulating and inhibiting the NF-κB and caspase pathways.

### 2.7. CHRs Inhibit the Expression of NF-κB, Caspase3, and Caspase9 by WB

Additionally, we measured the expression of proteins associated with inflammation and apoptosis via WB (Figure 6a). As shown in Figure 6, the level of p-NF-κB was significantly increased in the mice with gastric ulcers in the model group, but the CHR pretreatments inhibited the phosphorylated expression of NF-κB. Similarly, we found that Caspase3 and Caspase9 were significantly up-regulated in the model group, while their levels were attenuated by the CHR pretreatments; this is clearly shown in the quantitative analysis diagrams in Figure 6b. These results also suggest that CHRs exert significant anti-inflammatory and anti-apoptotic effects by down-regulating the NF-κB and caspase pathways.

### 2.8. Molecular Docking and Visualization

Molecular docking provides preliminary experimental data at the virtual level, which can provide a reference for further experiments and verify the mechanisms determined. Molecular docking was used to predict the binding affinity between core compounds and the targets of inflammation and apoptosis pathways. The molecular docking results indicated the presence of affinities between the two chromones and ABCB1, ALOX5, NF-κB, Caspase9, and Bcl-2, suggesting that the stable binding of the core compounds and key target played a role. These results are consistent with the mechanism of this action.

Molecular docking patterns were visualized for the obtained candidate compounds with a high affinity for specific targets (Figure 7a,b), showing their patterns of binding to amino acid residues at specific sites on the target. The main goal was to investigate the types of forces that formed, with the number of hydrogen bonds being the most important factor. Differences in the types of interactions and the amino acid residues involved in the interactions were further compared. Using molecular docking, the CHRs were docked to five proteins. In general, the binding energy was less than −5 kcal/mol, indicating that the receptor and ligand can bind together well. The lower the binding energy, the more tightly the receptor and ligand can bind. It can be clearly seen that the chromone compounds bind to five core proteins through hydrogen bonding and the van der Waals force, and the binding energies are all less than −5 kcal/mol; among them, the binding energy with ABCB1 is as high as −8.2 kcal/mol. The CHRs bound closely to the target proteins, indicating that these target proteins are key anti-gastric ulcer targets. This verified the conclusions of previous studies on target spectrum prediction and anti-inflammatory and anti-apoptotic mechanisms.

## 3. Discussion

Agarwood had been used to treat diseases, particularly gastric ulcers, for centuries. Our previous study revealed that agarwood extracts provide gastric mucosal protection [23,24]. This study found that the gastro-protective effect of CHRs on ethanol-induced gastric injury relieved the severity of tissue damage in gastric tissue, decreasing the secretion of inflammatory cytokines and down-regulating the protein expressions of anti-inflammatory and anti-apoptotic factors.

Excessive alcohol consumption is the greatest contributing factor to gastric ulcers and a common means of establishing gastric ulcer models. Gastric ulcers result in changes in the mucosal edema, hemorrhages, and inflammatory cell infiltration, showing similarity to the characteristics of human acute peptic ulcer disease [32,33]. In this study, we chose the model to represent the consequences of excessive drinking. Ulcer index values and histological lesions were used to measure gastric mucosal damage. Our study showed that anhydrous ethanol (0.15 mL/10 g) produced noticeable mucosal damage. Pretreatment with CHRs substantially reduced the ulcer index and the formation of histological lesions compared to a model group, suggesting that CHRs have a significant protective effect against gastric cell injury induced by alcohol.

Previous research showed that gastric ulcers caused by ethanol intake may trigger and activate the inflammatory system, accompanied by up-regulated levels of pro-inflammatory cytokines, such as TNF-α, IL-6, IL-12, and IL-18 [34]. Inflammation is another important mechanism involved in the pathogenesis of gastric ulcers. NF-κB is a well-known vital transcription factor in the acute phase of the inflammatory process, inducing the up-regulation of pro-inflammatory cytokines. When gastric mucosal cells receive exogenous stimuli from ethanol, NF-κB is activated and combines with signaling pathway upstream and downstream factors, such as ABCB1 and ALOX5, triggering the target inflammatory response and aggravating gastric mucosa damage [35,36]. Interestingly, CHRs potentially have anti-inflammatory effects on ethanol-induced gastric damage. Pretreatment with CHRs had a significant inhibitory effect on the pro-inflammatory cytokine elements of TNF-α, IL-6, IL-12, and IL-18 and mitigated gastric damage by inhibiting the NF-κB pathway and down-regulating the expressions of ABCB1, ALOX5, and p-NF-κB, suggesting that CHRs act against the inflammation of gastric ulcers. In fact, our previous study found that agarwood extract exerted pronounced anti-inflammatory capabilities in a variety of disease models [37,38], which corroborates our present results.

This investigation revealed that apoptosis also plays an important role in the occurrence and development of gastric ulcers, especially in the caspase family signaling pathway [39,40]. Mechanistically, evidence indicates that ethanol intake down-regulates the anti-apoptosis gene Bcl-2, an effect markedly driven by the accumulation of oxygen-derived free radicals and pro-inflammatory signals in the area of the ulcer area. These circumstances result in the restriction of the apoptotic protein Bax, which, in turn, leads to the mitochondrial escape of cytochrome C, subsequently activating Caspase 9, and finally, Caspase3, resulting in cell death [41,42]. The CHRs pretreatment clearly down-regulated the expression of Caspase 3 and Caspase 9 in mice with ethanol-induced gastric ulcers. In contrast, their Bcl-2 levels were further remarkably up-regulated by the CHR pretreatment in a dose-dependent manner. Our data suggest that pretreatment with CHRs might inhibit the apoptosis of gastric mucosal cells, contributing to the anti-apoptotic strategy against gastric ulcer damage. Previous studies have demonstrated latent anti-apoptotic activity [38,43].

## 4. Materials and Methods

### 4.1. Materials and Reagents

2-(2-Phenylethyl)chromone (CHR1) and 2-(2-(4-methoxyphenyI)ethyl)chromone (CHR2) (reagent grade, purity ≥ 95%) were purchased from Shanghai Macklin Biochemical Technology Co., Ltd., (Shanghai, China). Omeprazole enteric-coated capsules (OEC) were provided by Zibo Wanjie Pharmaceutical Co., Ltd. (Zibo, China). Enzyme-linked immunosorbent assay (ELISA) kits for determination of cytokines (IL-6, IL-12, IL-18, and TNF-α) were produced by Coaibo Biotechnology Co., Ltd. (Shanghai, China). Anhydrous ethanol and other chemicals procured were of analytical grade.

### 4.2. Prediction of Disease Spectrum and Target of Agarwood Chromone via Network Pharmacology

Agarwood CHR1 and CHR2, described in Table 2 and Figure 8, were systematically obtained to construct the chromone composition-disease spectrum and target by network pharmacological methods, screen and label the genes related to apoptosis and inflammation, and predict the discovery of two chromone compounds closely related to apoptosis and inflammation.

In order to obtain potential target spectrum data of chromone, 10 data sources listed were used in this study for target data acquisition and prediction. Finally, the UniProt database (http://www.uniprot.org, 6 September 2022) was used to standardize the gene names of the target data, and only *Homo sapiens* target data were retained for subsequent analysis.

### 4.3. Construction of Inflammatory and Apoptotic Gene Sets

The set of genes related to inflammation and apoptosis were searched and extracted from the database. The keywords relating to inflammation were “inflammation”, “inflammatory”, “inflammatory pathway”, and “inflammatory response”. Apoptosis was searched in the Gene Cards database using the keywords “Apoptosis”, “Apoptosis pathway”, and “Apoptotic pathway”. The results obtained using different search terms were integrated, and their frequency was calculated. The higher the Total Relevance score, the higher the reliability of data derived from the multiple search terms.

### 4.4. Animals and Experimental Procedure

Male ICR mice (20 ± 2 g, 4–6 weeks) were purchased from the Hainan Institute of Materia Medica (Haikou, China). The animals were maintained in a 12 h light/dark cycle at a temperature of 20–24 °C and 50–70% humidity. They were housed for three days prior to the experiments. Animal care and experimental protocols were approved by the Animal Care and Use Committee at the Hainan Institute of Materia Medica.

Seventy mice were randomly divided into seven groups. The groups were as follows: normal control group, model group, OEC group, CHR1 group, and CHR2 group. The normal and model groups were orally administered 20 mL/kg of distilled water. The OEC group was pretreated with 40 mg/kg daily via oral garage. The CHRs were administered at 20 and 40 mg/kg doses. After pretreatment for 7 days and 12 h of food deprivation with water ad libitum, the other mice were used to develop an acute gastric ulcer model by administering absolute ethanol at a concentration of 0.15 mL/10 g via oral gavage. The normal group was excluded. Whole blood samples were collected from the orbit for serum analysis 1 h after the administration of absolute ethanol, and the animals were then sacrificed via cervical dislocation. Their stomachs were immediately removed, fixed with a formaldehyde solution, and prepared for the determination of the gastric lesion index, histopathological sections, and immunohistochemical analysis.

### 4.5. Determination of Ulcer Index

The stomachs were fixed in a 4% formaldehyde solution for 1h. The degree of gastric mucosal defects in both the inner and outer layer was observed. The length (mm) of each mucosal lesion was measured with visual inspection. The score was estimated as follows: a blood point warrants 1 score, a line length less than 1 mm is worth 2 scores, a line length of 1–2 mm is worth 3 scores, a line length of 3–4 mm is worth 4 scores, and line lengths of 5 mm and longer are worth 5 scores [44]. The total score for the whole stomach was the ulcer index. We also calculated the ulcer inhibition rate. The percentage ulcer inhibition rate was calculated as follows:Ulcer inhibition rate (%) = (Mean ulcer indexes of model group-Mean ulcer indexes of treatment group)/Mean ulcer indexes of model group × 100

### 4.6. Tissue Damage Evaluation

Stomach tissues were removed and fixed in a formaldehyde solution, dehydrated, embedded in paraffin and sectioned. For tissue damage analysis, the samples were sectioned at 5 µm and stained with hematoxylin-eosin (HE) using a standard operation. They were then observed under a microscope at 400× magnification and photographed. Tissue damage was calculated using the following score system (LDI score): a 1 cm segment of each tissue damage section was assessed for epithelial cell loss (score: 0–3), edema in the upper mucosa (score: 0–4), hemorrhagic damage (score: 0–4), and the presence of inflammatory cells (score: 0–3) [45].

### 4.7. Detection of IL-6, IL-12, IL-18, and TNF-α in Serum

Serum samples were collected to assess the effects of the agarwood extracts on mouse stomach tissue. The cytokines of IL-6, IL-12, IL-18, and TNF-α in the serum were measured using mouse-specific ELISA kits according to the manufacturer’s instructions.

### 4.8. Detection of Protein Expression via Immunohistochemistry (IHC)

The stomach tissues were fixed in a formalin solution, dehydrated with ethanol, embedded in paraffin, and sectioned. Then, 5 µm sections were sealed with a buffered blocking solution in phosphate-buffered saline (PBS) (containing 3% bovine serum albumin) for 15 min. The primary antibodies used were as follows: ABCB1 (sc-13131), ALOX5 (sc-136195), NF-κB (sc-51682), Caspase9 (sc-56076), and Bcl-2 (sc-7382), which were purchased from Santa Cruz Biotech (Santa Cruz, CA, USA). The sections were then co-incubated with the primary antibody for ABCB1, ALOX5, NF-κB, Caspase9, and Bcl-2 at a dilution of 1:50 in PBS (*v*/*v*), at 4 °C overnight. The sections were then washed with PBS and incubated with the HRP-coupled secondary antibody (Cell Signaling, Danvers, MA, USA) at a dilution of 1:500 at room temperature for 1 h. The sections were then washed with 0.05 M of Tris-HCl at a pH of 7.66, and co-incubated with a 3, 30-diaminobenzidine solution in the dark, at room temperature for 10 min. The sections were washed again, stained with hematoxylin and observed under a light microscope. Image-ProPlus 6.0 Software (Media Cybernetics, Rockville, MD, USA) was used to analyze protein expression.

### 4.9. Detection of Protein Expression via Western Blotting (WB)

The stomach tissues were homogenized in a standard RIPA buffer supplemented with a cocktail of protease and phosphatase inhibitors. The homogenate was then centrifuged at 15,000× *g* at 4 °C for 10 min. The protein concentration was determined using a BCA Protein Assay Kit (Beyotime, P0010, Shanghai, China). According to the molecular weight of the target protein, the appropriate concentration of separation gel was selected, the sample volume was brought to 20 μL and electrophoresed, and the proteins were transferred to a PVDF membrane and blocked with 5% skim milk powder for 1 h. The primary antibodies used were as follows: Phospho-NF-κB p65 (Cell signaling, 3033S, Danvers, MA, USA), Caspase3 (Cell signaling, 9662S), Caspase9 (Cell signaling, 9504T), and β-actin (Beyotime, AF0003, Shanghai, China). The primary antibodies against β-actin (1:1000), NF-κB, Caspase3, and Caspase9 (1:500) were added. The membranes were incubated overnight at 4 °C, and the HRP-coupled secondary antibody (Cell Signaling, 1:2000) was incubated with the membrane at room temperature for 2 h. ECL chemiluminescence was performed with a gel imager. Gel-Pro Analyzer 7.0 was used for grayscale scanning and quantitative analysis, and intuitive histograms were produced.

### 4.10. Molecular Docking Verification and Visualization

Combined with the results of the network pharmacological prediction and the mechanism of action, the affinity and visualization of the binding between the active compound and the core target were deeply analyzed via computer virtual design and molecular docking analysis to verify the mechanism of the anti-gastric ulcer effect of the agarwood chromone compound.

Molecular docking was used to predict the binding affinity between the chromone compounds and five targets of inflammation and apoptosis pathways, which provided a reference for further experimental verification. AutoDock Vina 1.1.2 is an open-source program for molecular docking and virtual screening, that provides a high average accuracy of binding mode predictions for docking experiments [46]. Compounds were downloaded from the PubChem database (https://pubchem.ncbi.nlm.nih.gov/, 6 September 2022) [47] as SDF files and converted through Open Babel against 2.4.1 into MOL2 format. The three-dimensional structure of the target proteins was obtained from the Protein Database (PDB) (https://www.rcsb.org/, 6 September 2022) [48]. For detailed PDB_id information, see the following table. Ligands and receptors were prepared using the AutoDock Vina 1.1.2 tutorial. For each structure, the water molecules were removed, non-polar hydrogen was added, and the Gasteiger charge was calculated, and the structure saved in PDBQT format. Generally, the threshold value is Affinity ≤ −5 kcal/mol. The lower the Vina score is, the higher the affinity between the components and proteins is. Usually, the lowest affinity is regarded as the best docking value and the visual interaction mode is shown by PLIP [49].

### 4.11. Statistical Analysis

Data were expressed as mean ± SD values for ten animals in each group and statistically evaluated with SPSS 17.0 software (IBM Corporation, New York, NY, USA). Differences between each group were analyzed using one-way ANOVA and Tukey’s post hoc test. *p* < 0.05 was considered statistically significant.

## 5. Conclusions

The results of this study clearly demonstrated that CHRs exerted protective effects against ethanol-induced gastric ulcers. Their potential mechanisms of action are related to anti-inflammatory and anti-apoptosis activities via the inhibition of the NF-κB and caspase family pathways, but how they affect the function of upstream and downstream signaling molecules in the entire pathway remains unclear. Both of the chromone compounds showed a good anti-ulcer effect, although the effect of CHR2 was relatively stronger, possibly because the CHR2 compound contains more active methoxy groups, which better facilitated its binding to the target protein. This point is worth further study. Chromone and its derivatives are abundant compounds in agarwood. They exhibit significant activity and have great potential to be developed into innovative drugs for the prevention and treatment of chronic inflammatory diseases, such as gastrointestinal illnesses. This study also produced diagrams of the molecular mechanism of the anti-gastric ulcer effect of CHRs (Figure 9). As a material used in traditional Chinese medicine, agarwood has “multi-component, multi-target and multi-pathway” characteristics, and its anti-gastric ulcer mechanism has not been fully explained, necessitating further investigations into the anti-gastric ulcer effect and the specific mechanism of agarwood chromone, its derivatives and other active ingredients. Moreover, the pathogenesis of gastric ulcer is complex and diverse; in addition to inflammation and apoptosis, there are many contributing factors, such as the excessive secretion of gastric acid, weakened gastric mucosal barrier function, *Helicobacter pylori* infection, and the excessive use of non-steroidal anti-inflammatory drugs. In future studies, researchers can consider the influence of agarwood and its monomeric compounds on the above factors to further explore its anti-gastric ulcer mechanism. These monomeric compounds, CHRs, offer a good protective effect against gastric ulcers, providing a reference for the development of new monomeric drugs from agarwood to be used in treatment of gastric ulcers. In summary, this study indicates that agarwood chromone could be developed into a potential protective drug for the prevention and treatment of gastric ulcers.

## Figures and Tables

**Figure 1 pharmaceuticals-18-00514-f001:**
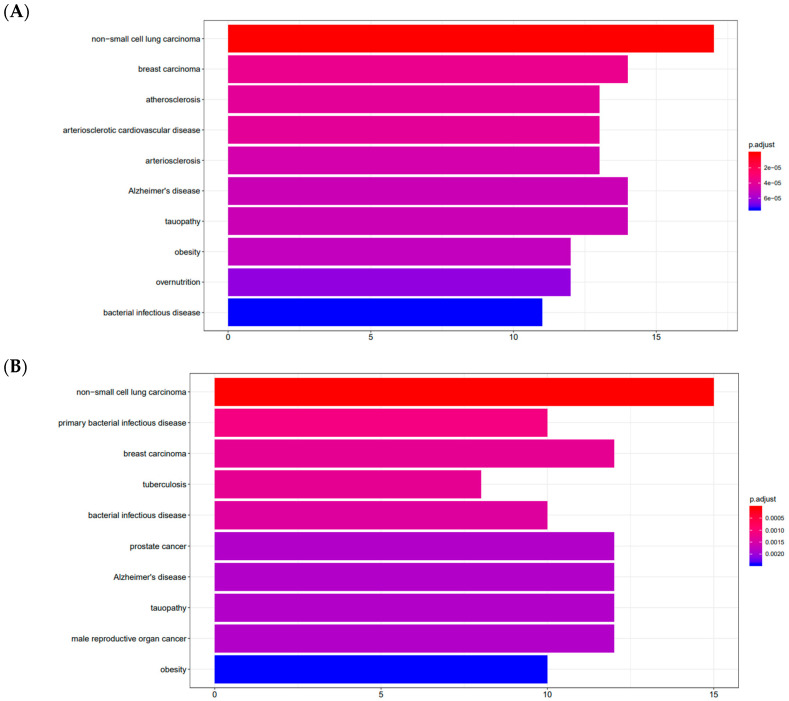
Disease spectra and mechanism signaling pathways of CHRs predicted via network pharmacology. (**A**) CHR1 disease spectrum; (**B**) CHR1 mechanism pathway; (**C**) CHR2 disease spectrum; (**D**) CHR2 mechanism pathway.

**Figure 2 pharmaceuticals-18-00514-f002:**
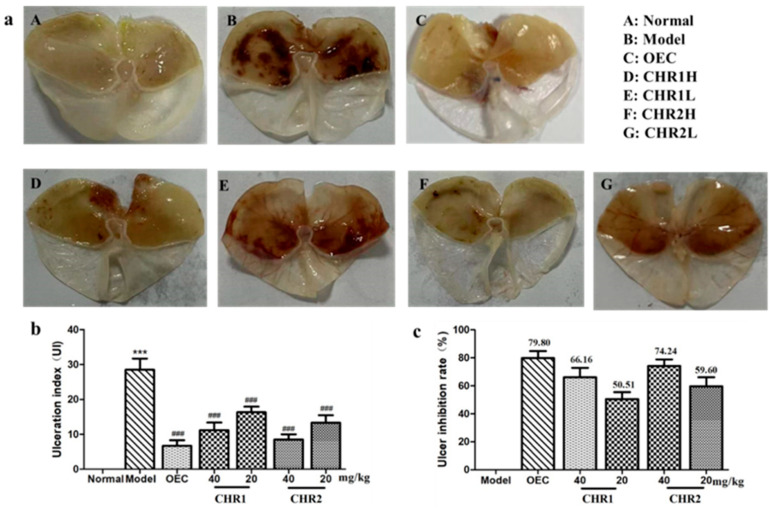
Effect of CHRs on gastric ulcer induced using anhydrous ethanol in mice. Animals were treated with OEC (40 mg/kg), CHR1H (40 mg/kg), CHR1L (20 mg/kg), CHR2H (40 mg/kg) or CHR2L (20 mg/kg) via oral administration. Normal and model groups were given distilled water. (**a**) Typical pictures of stomachs; (**b**) gastric lesions index; (**c**) ulcer inhibition rate. Data are expressed as mean ± SD (*n* = 6). *** *p* < 0.001 vs. normal group; ### *p* < 0.001 vs. model group.

**Figure 3 pharmaceuticals-18-00514-f003:**
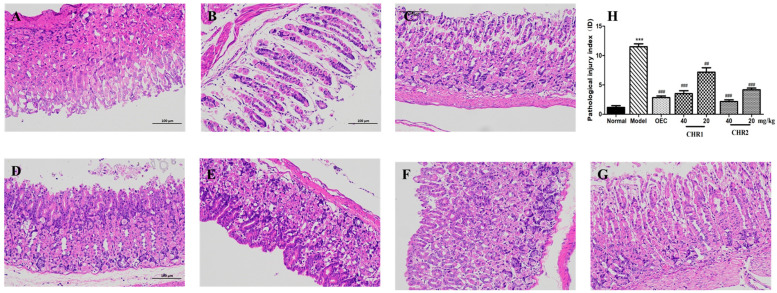
Effect of CHRs on gastric histopathology, staining with hematoxylin and eosin (HE), scale bar = 100 µm. Animals were treated with OEC (40 mg/kg), CHR1H (40 mg/kg), CHR1L (20 mg/kg), CHR2H (40 mg/kg), and CHR2L (20 mg/kg) via oral administration. Normal and model groups were given distilled water. (**A**) Normal; (**B**) model; (**C**) OEC (40 mg/kg); (**D**) CHR1H (40 mg/kg); (**E**) CHR1L (20 mg/kg); (**F**) CHR2H (40 mg/kg); (**G**) CHR2L (20 mg/kg), and (**H**) Pathological damage scores analysis from HE staining. Data are expressed as mean ± SD (*n* = 4). *** *p* < 0.001 vs. normal group; ## *p* < 0.01, ### *p* < 0.001 vs. model group.

**Figure 4 pharmaceuticals-18-00514-f004:**
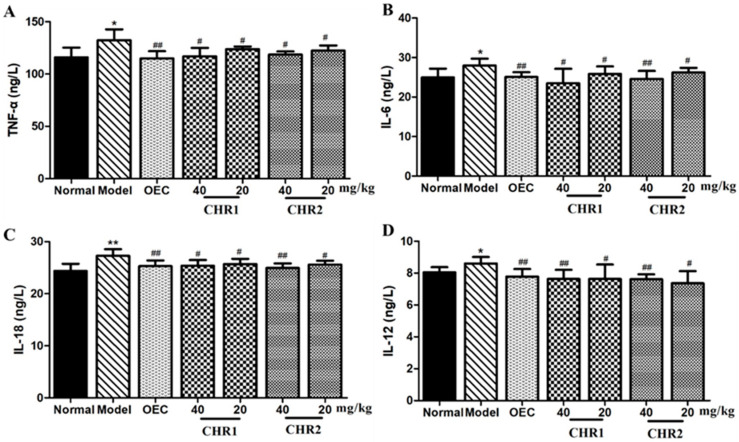
Effect of CHRs on levels of inflammatory cytokines in mouse serum. (**A**) TNF-α, (**B**) IL-6, (**C**) IL-18, and (**D**) IL-12. Data are expressed as mean ± SD (*n* = 6). * *p* < 0.05, ** *p* < 0.01 vs. normal group; # *p* < 0.05, ## *p* < 0.01 vs. model group.

**Figure 5 pharmaceuticals-18-00514-f005:**
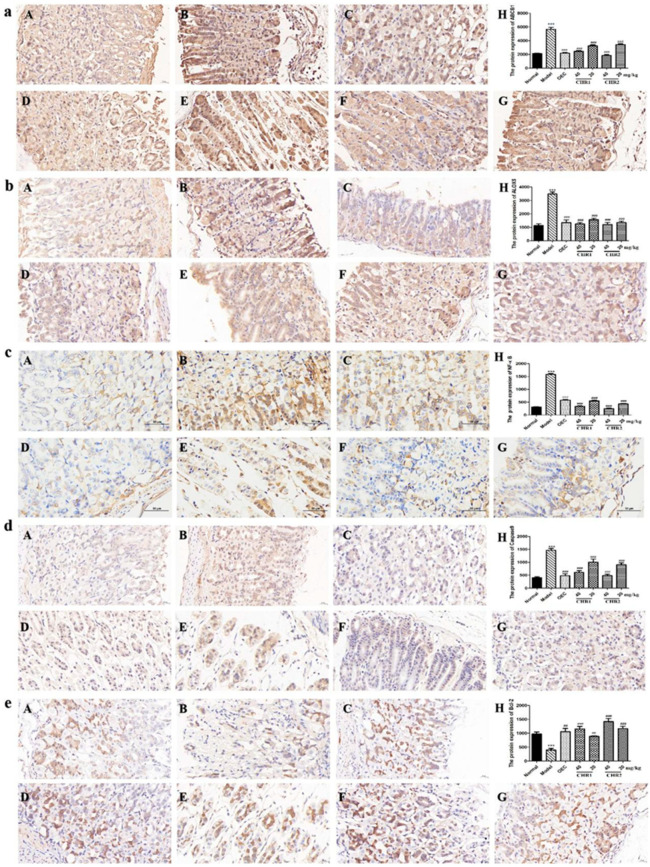
Effect of CHRs on the protein expression of ABCB1, ALOX5, NF-κB, Caspase9, and Bcl-2, scale bar = 20 µm. Administration and grouping are shown in Figure 4. (**a**) Protein expression of ABCB1; (**b**) protein expression of ALOX5; (**c**) protein expression of NF-κB; (**d**) protein expression of Caspase9, and (**e**) protein expression of Bcl-2. (A) Normal; (B) Model; (C) OEC (40 mg/kg); (D) CHR1H (40 mg/kg); (E) CHR1L (20 mg/kg); (F) CHR2H (40 mg/kg); (G) CHR2L (20 mg/kg), and (H) Quantization of protein expression. Data are expressed as mean ± SD (*n* = 3). *** *p* < 0.001 vs. normal group; ## *p* < 0.01, ### *p* < 0.001 vs. model group.

**Figure 6 pharmaceuticals-18-00514-f006:**
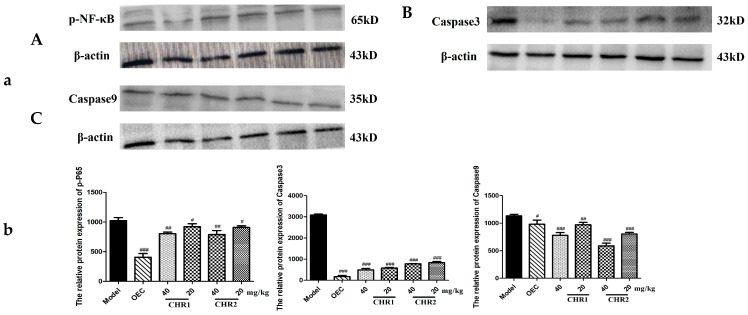
Effect of CHRs on protein expression of p-NF-κB, Caspase3 and Caspase9. Administration and grouping are shown in this figure. (**A**) Protein expression of p-NF-κB; (**B**) Protein expression of Caspase3; (**C**) Protein expression of Caspase9. (**a**) Target protein and reference protein banding map; (**b**) quantification of protein expression. Data are expressed as mean ± SD (*n* = 3). # *p* < 0.05, ## *p* < 0.01, ### *p* < 0.001 vs. model group.

**Figure 7 pharmaceuticals-18-00514-f007:**
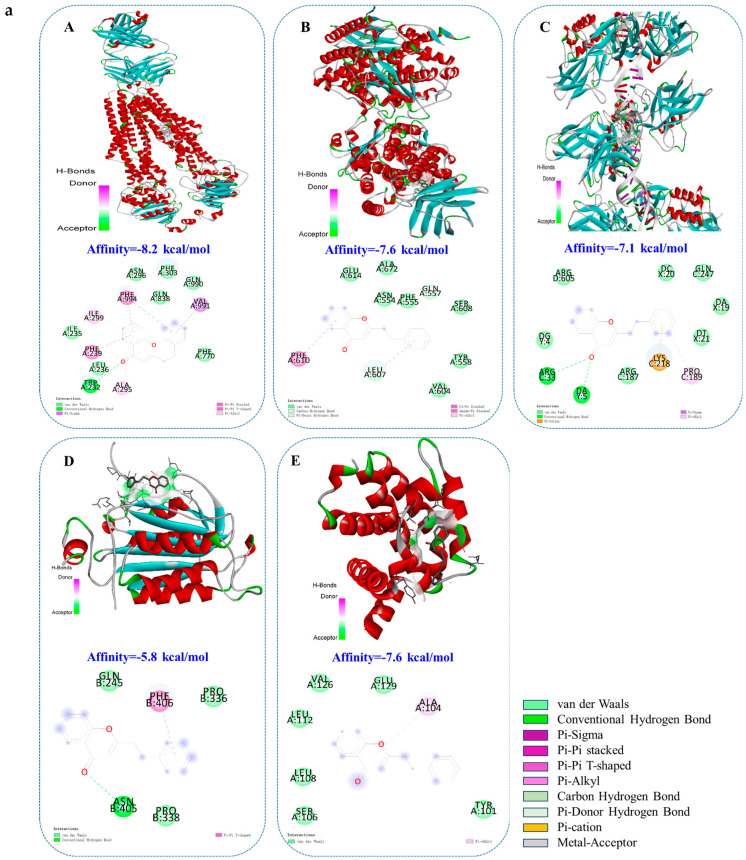
Molecular docking and visualization of CHRs and (**A**): ABCB1, (**B**): ALOX5, (**C**): NF-κB, (**D**): Caspase9 and (**E**): Bcl-2. (**a**) CHR1; (**b**) CHR2.

**Figure 8 pharmaceuticals-18-00514-f008:**
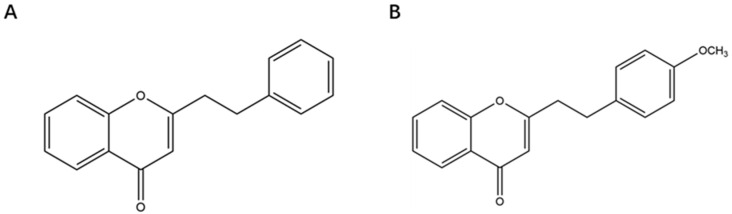
The structural formula for the chromone compounds, (**A**): 2-(2-phenylethyl)chromone; (**B**): 2-(2-(4-methoxyphenyI)ethyl)chromone.

**Figure 9 pharmaceuticals-18-00514-f009:**
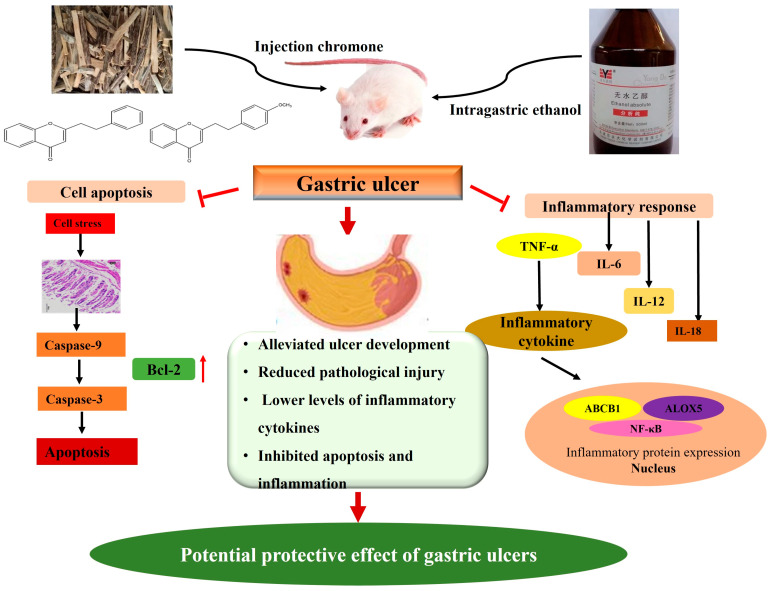
Mechanism diagram of anti-gastric ulcer effect of CHRs.

**Table 1 pharmaceuticals-18-00514-t001:** Five targets of inflammation and apoptosis pathways.

Name	Gene Name	PDB_id	Gene Symbol
ABCB1	ABCB1	7a69	ATP binding box transporter B1
ALOX5	ALOX5	3v99	arachidonic acid-5-lipoxygenase
NF-κB	NFKB1	3gut	nuclear factor of kappa light polypeptide gene enhancer in B cells 1
Bcl-2	BCL2	3sp7	B cell leukemia/lymphoma 2
Caspase9	CASP9	1nw9	caspase 9

**Table 2 pharmaceuticals-18-00514-t002:** Basic Information of chromone compounds.

CID	Compound Name	Molecular Formula	Molecular Weight	CHEMBL_ID
441,964	2-(2-phenylethyl)chromone	C17H14O2	250.29	CHEMBL481060
185,208	2-(2-(4-methoxyphenyI)ethyl)chromone	C18H16O3	280.3	CHEMBL4468180

## Data Availability

Materials and data are available from the first author or corresponding author. Data is contained within the article.

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
