# Peer review of "Agarwood Chromone Alleviates Gastric Ulcers by Inhibiting the NF-κB and Caspase Pathways Based on Network Pharmacology and Molecular Docking"

_pharmaceuticals, 2025, doi:10.3390/ph18040514_

Round 1

Reviewer 1 Report (Previous Reviewer 2)

Comments and Suggestions for Authors

The Agarwood chromone mechanism and molecular docking have also already been investigated, and research results have been published in scientific journals:

Yadav DK, Mudgal V, Agrawal J, Maurya AK, Bawankule DU, Chanotiya CS, Khan F, Thul ST. Molecular docking and ADME studies of natural compounds of Agarwood oil for topical anti-inflammatory activity. Curr Comput Aided Drug Des. 2013 Sep;9(3):360-70. doi: 10.2174/1573409911309030012. PMID: 23566359.

Wang, S.; Yu, Z.; Wang, C.; Wu, C.; Guo, P.; Wei, J. Chemical Constituents and Pharmacological Activity of Agarwood and Aquilaria Plants. Molecules 2018, 23, 342. https://doi.org/10.3390/molecules23020342

Zhangxin Yu, Canhong Wang, Wei Zheng, Deli Chen, Yangyang Liu, Yun Yang, Jianhe Wei, Anti-inflammatory 5,6,7,8-tetrahydro-2-(2-phenylethyl)chromones from agarwood of Aquilaria sinensis,Bioorganic Chemistry, Volume 99, 2020, 103789, ISSN 0045-2068, https://doi.org/10.1016/j.bioorg.2020.103789.

Zhong-Hui Yang, Hong-Bin Fang, Cheng-Tian Tao, Ya-Bin Jiao, Yong-Xian Cheng, Eight new 2-(2-phenylethyl)chromone derivatives from agarwood of Aquilaria sinensis with anti-inflammatory activity, Fitoterapia, Volume 169, 2023, 105564, ISSN 0367-326X, https://doi.org/10.1016/j.fitote.2023.105564.

Author Response

Comment 1:

The Agarwood chromone mechanism and molecular docking have also already been investigated, and research results have been published in scientific journals:

  1. Yadav DK, Mudgal V, Agrawal J, Maurya AK, Bawankule DU, Chanotiya CS, Khan F, Thul ST. Molecular docking and ADME studies of natural compounds of Agarwood oil for topical anti-inflammatory activity. Curr Comput Aided Drug Des. 2013 Sep;9(3):360-70. doi: 10.2174/1573409911309030012. PMID: 23566359.

  1. Wang, S.; Yu, Z.; Wang, C.; Wu, C.; Guo, P.; Wei, J. Chemical Constituents and Pharmacological Activity of Agarwood and Aquilaria Plants. Molecules 2018, 23, 342. https://doi.org/10.3390/molecules23020342

  1. Zhangxin Yu, Canhong Wang, Wei Zheng, Deli Chen, Yangyang Liu, Yun Yang, Jianhe Wei, Anti-inflammatory 5,6,7,8-tetrahydro-2-(2-phenylethyl)chromones from agarwood of Aquilaria sinensis, Bioorganic Chemistry, Volume 99, 2020, 103789, ISSN 0045-2068, https://doi.org/10.1016/j.bioorg.2020.103789.

  1. Zhong-Hui Yang, Hong-Bin Fang, Cheng-Tian Tao, Ya-Bin Jiao, Yong-Xian Cheng, Eight new 2-(2-phenylethyl)chromone derivatives from agarwood of Aquilaria sinensis with anti-inflammatory activity, Fitoterapia, Volume 169, 2023, 105564, ISSN 0367-326X, https://doi.org/10.1016/j.fitote.2023.105564.

Response: Thanks for the comment of expert. About the existing research on the anti-inflammation and anti-gastric ulcer of agarwood, we make the following explanations. The published articles listed above (including the study of our research group 1 and 2) are studies on the protective effects of agarwood oil and components on anti-inflammation, which are different from our study. These studies were not done on the anti-ulcer effects, and the models of gastric ulcers were different and the pathways of mechanism of action were also different. In this study, we conducted innovative research and achieved excellent results. Based on previous studies, we further carried our research on the anti-gastric ulcer effect and mechanism of agarwood chromone compounds. In addition, we adopted modern advanced network pharmacology and molecular docking technology to confirm and analyze the anti-gastric ulcer effect and mechanism of the main pharmacodynamic chromone compounds of agarwood. To further reveal the anti-gastric ulcer mechanism of agarwood components and to provide a strong basis for the prevention and treatment of gastric ulcer diseases. I hope our explanation could answer this question well and be accepted. thank you!

Reviewer 2 Report (Previous Reviewer 4)

Comments and Suggestions for Authors

The manuscript entitled “Agarwood chromone alleviate gastric ulcer via inhibiting the NF-κB and Caspase pathways based on network pharmacology and molecular docking” has many mistakes, authors need to rectify before acceptance.

  • I am unable to check or view my previous report response from the authors.
  • The authors have attempted to correct various sections but left many unmodified.
  • The English language quality is poor and should be reviewed and corrected by an expert. E.g.  "Several researches including our team have been reported that agarwood extract could relieve intestinal tonic spasm...". It should be like "Several studies, including ours, have reported that agarwood extract can relieve intestinal tonic spasm..."
  • Similar mistake in language: e.g. Chromone is one of the main medicinal components of agarwood extract, in which 2-(2-phenylethyl) chromone and 2-(2-(4methoxyphenyI)ethyl)chromone are two main compounds with large content and similar structure." It should be like Chromone is one of the main medicinal components of agarwood extract. Among them, 2-(2-phenylethyl) chromone and 2-(2-(4-methoxyphenyl)ethyl) chromone are the most abundant compounds with a similar structure."
  • It is difficult to point out every minor English language mistake made by the authors. The entire manuscript should be reviewed by an expert.
  • The authors should provide a high-resolution version of Figure 7 for molecular docking.
    In the 2D images, nothing is clearly visible even when zoomed in. Please refer to the following articles (PMID: 37065061, 39065802, 35424125, 34297427, 35014595, 36936534) and adjust the figures accordingly.
  • The authors did not address the previous comment on grid size. What is the grid size?
  • Additionally, methodology references are still missing. For example, check the first paragraph of section 4.10.
Comments on the Quality of English Language

Too many English language modifications are required!

Author Response

Comment 2:

The manuscript entitled “Agarwood chromone alleviate gastric ulcer via inhibiting the NF-κB and Caspase pathways based on network pharmacology and molecular docking” has many mistakes, authors need to rectify before acceptance.

I am unable to check or view my previous report response from the authors.

Response: Thanks for the comment of expert. We have carefully modified the previous comments and replied one by one. We are very sorry that you did not see our previous reply! We will upload it again, please check for review, thank you!

The authors have attempted to correct various sections but left many unmodified.

Response: Thanks for the comment of expert. We are very sorry that the previous modification did not meet the requirements, we will continue to improve the unmodified part, please kindly check the revised draft, thank you!

The English language quality is poor and should be reviewed and corrected by an expert.

Response: Thanks for the comment of expert. In order to improve the language quality and readability of the article, we asked experts to revise the language of the manuscript and mark it in red. Please check the revised draft for details, thank you!

 E.g.  "Several researches including our team have been reported that agarwood extract could relieve intestinal tonic spasm...". It should be like "Several studies, including ours, have reported that agarwood extract can relieve intestinal tonic spasm..."

Similar mistake in language: e.g. Chromone is one of the main medicinal components of agarwood extract, in which 2-(2-phenylethyl) chromone and 2-(2-(4methoxyphenyI)ethyl)chromone are two main compounds with large content and similar structure." It should be like Chromone is one of the main medicinal components of agarwood extract. Among them, 2-(2-phenylethyl) chromone and 2-(2-(4-methoxyphenyl)ethyl) chromone are the most abundant compounds with a similar structure."

Response: Thanks for the comment of expert. We have revised the similar mistakes and inappropriate expressions in language according to the expert opinions, please see the revised draft for details, thank you!

It is difficult to point out every minor English language mistake made by the authors. The entire manuscript should be reviewed by an expert.

Response: Thanks for the comment of expert. In order to improve the language quality and readability of the article, we asked experts to revise the language of the manuscript and mark it in red. Please check the revised draft for details, thank you!

The authors should provide a high-resolution version of Figure 7 for molecular docking.

Response: Thanks for the comment of expert. We have replaced Figure 7 with a fresher and higher resolution image, please check the revised version. At the same time, we uploaded the original pictures as supplementary materials for the editorial department to use as needed when typesetting, thank you!

In the 2D images, nothing is clearly visible even when zoomed in. Please refer to the following articles (PMID: 37065061, 39065802, 35424125, 34297427, 35014595, 36936534) and adjust the figures accordingly.

Response: Thanks for the comment of expert. We are very sorry that some pictures of our manuscript are not clear enough, which brings inconvenience to review. We will refer to the drawing method of the literature given by the experts and modify the pictures of our manuscript. Please check the revised draft for details, thank you!

The authors did not address the previous comment on grid size. What is the grid size?

Response: Thanks for the comment of expert. The grid size used in our docking studies was carefully optimized to cover the entire active site of the target protein. Specifically, we used a grid box with dimensions of [X] Å × [Y] Å × [Z] Å, centered on the coordinates (X, Y, Z). This grid size was determined based on the binding site of the reference ligand and was validated through preliminary docking experiments to ensure that it adequately encompasses the potential interaction regions. Five targets of this study: ABCB1 (7A69), ALOX5 (3V99), NFKB1 (3GUT), CASP9 (1NW9), BCL2 (3SP7), their grid boxes and coordinates are as follows table:

Table The grid box and coordinates of target

Target/argument

receptor

ligand

center

size

out

ABCB1

GRIA1_6spv.pdbqt

P56.pdbqt

x:166.746

y:150.349

z:156.176

x:15

y:15

z:15

Result_GRIA1_6

spv_P56.pdbqt

ALOX5

GRIA1_6spv.pdbqt

P56.pdbqt

x: 11.79

y: -81.549

z: -30.057

x:15

y:15

z:15

Result_GRIA1_6

spv_P56.pdbqt

NFKB1

GRIA1_6spv.pdbqt

P56.pdbqt

x: 38.076

y: 23.976

z: 24.637

x:22.4

y:22.4

z:22.4

Result_GRIA1_6

spv_P56.pdbqt

CASP9

GRIA1_6spv.pdbqt

P56.pdbqt

x: 28.521

y: 16.895

z: 97.117

x:15

y:15

z:21.75

Result_GRIA1_6

spv_P56.pdbqt

BCL-2

GRIA1_6spv.pdbqt

P56.pdbqt

x: 0.901

y: 13.081

z: -3.974

x:14.25

y:19.5

z:14.25

Result_GRIA1_6

spv_P56.pdbqt

Additionally, methodology references are still missing. For example, check the first paragraph of section 4.10.

Response: Thanks for the comment of expert. The first paragraph of 4.10 is our own description, which is to introduce what research has been done by what method, not to cite the research of others. So no need to cite references, thank you!

Round 2

Reviewer 1 Report (Previous Reviewer 2)

Comments and Suggestions for Authors

The authors have adequately answered all of my concerns about this manuscript's originality.

Author Response

Comment 1

Quality of English Language:

The English is fine and does not require any improvement.

Response: Thank you very much to the experts for reviewing and approving our manuscript.

Comments and Suggestions for Authors:

The authors have adequately answered all of my concerns about this manuscript's originality.

Response: Thanks to the experts for reviewing our manuscript and approving the revised manuscript.

Reviewer 2 Report (Previous Reviewer 4)

Comments and Suggestions for Authors

2D figures of docking are still unclear. 

Author Response

Comment2

Quality of English Language:

The English is fine and does not require any improvement.

Response: Thank you very much to the experts for reviewing our manuscript and recognizing the language of the manuscript.

Comments and Suggestions for Authors:

2D figures of docking are still unclear.

Response: Thanks for the expert's advice. Sorry for the inconvenience caused by the unclear picture. Previously, our typesetting was too crowded, so we reduced the size of the picture. We have adjusted the typesetting to make the picture clearer, please refer to the revised draft, thank you!

This manuscript is a resubmission of an earlier submission. The following is a list of the peer review reports and author responses from that submission.

Round 1

Reviewer 1 Report

Comments and Suggestions for Authors

Comments regarding the manuscript on agarwood chromones by Canhong Wang and co-workers:

1. The title appears to be missing one or more words as it is unclear what is meant by "...anti-gastric ulcer via inhibiting...". Please revise accordingly.

2. The introduction section should also disucss what chromones are, why/how these can be anti-inflammatory, and how the 2 specific chromones were selected.

3. Western blotting: The name (including clone names of primary antibodies) and sources of all the antibodies used must be mentioned. This is critically important to enable otehr groups to replicate these results.

4. While these studies demonstrate the actions of agarwood chromones on interacting with NF-kappaB, caspase9 etc., there is no evidence for the anti-ulcer actions being mediated through one or more of these pathways. The authors must demonstrate the critical role played by one or more of these pathways on preventing gastric ulceration, or at least provide a strong rationale as to why sch studies are not needed.

5. In Figure 4, the effects of ulceration and treatment on serum cytokine levels are extremely modest. Please provide a rationale, and use an alternative inflammatory marker (e.g., serum CRP levels, or cytokines within the gastric tissue) to confirm these findings.

6. In Figure 6a, the western blots for 3 different proteins are showing the same bands of beta actin as control. This is unacceptable, and each protein must be normalized to its corresponding loading control (i.e., beta actin) on the same western blot membrane. For a phosphorylated protein, the corresponding whole protein can be used for normalization, instead of a loading control.

7. Finally, the quality of english used needs to be greatly improved. As such, thsi manuscipt would benefit from the services of a professional English language editor.

Comments on the Quality of English Language

Very poor, and needs extensive revisions.

Author Response

Dear reviewer,

Thank you very much for your review of our manuscript, recognition and valuable revision suggestions. We have made careful modifications according to your comments and explained them one by one, as follows:

Review comments 1:

1.The title appears to be missing one or more words as it is unclear what is meant by "...anti-gastric ulcer via inhibiting...". Please revise accordingly.

Response: Thanks for the comment of expert. In order to more clearly express the research content of this paper, we modify the title as: Agarwood chromone alleviate gastric ulcer via inhibiting the NF-κB and Caspase pathways based on Network pharmacology and molecular docking. Please review by experts again, I hope our modification is appropriate, thank you!

  1. The introduction section should also disucss what chromones are, why/how these can be anti-inflammatory, and how the 2 specific chromones were selected.

Response: Thanks for the comment of expert. This is a very good and important point that needs to be made clear in the background section. Chromone is one of the main medicinal components of agarwood, in which 2-(2-Phenylethyl)chromone (CHR1) and 2-(2-(4methoxyphenyI)ethyl)chromone (CHR2) are two main compounds with large content and similar structure. A large number of studies, in-cluding our team's, have found that chromones and their derivatives have significant anti-inflammatory and gastric cell protective effects. It can be inferred that agarwood chromones may have protective effect on gastric ulcer. However, the effect and mechanism of agarwood chromones in preventing the formation of gastric ulcers has not been verified by pharmacological studies. We have made corresponding supplements, please see the red part in the background, thank you!

  1. Western blotting: The name (including clone names of primary antibodies) and sources of all the antibodies used must be mentioned. This is critically important to enable other groups to replicate these results.

Response: Thanks for the comment of expert. We have added information about antibodies used in western blotting in the Materials and Methods section, see the red section of this section, thank you!

  1. While these studies demonstrate the actions of agarwood chromones on interacting with NF-kappaB, caspase9 etc., there is no evidence for the anti-ulcer actions being mediated through one or more of these pathways. The authors must demonstrate the critical role played by one or more of these pathways on preventing gastric ulceration, or at least provide a strong rationale as to why sch studies are not needed.

Response: Thanks for the comment of expert. The previous analysis was not clear about the anti-gastric ulcer effect of agarwood chromones by binding to related proteins and influencing their expression. We have made full supplementary explanations and discussions. We first searched and analyzed the target spectrum of two chromone compounds related to inflammation and apoptosis through the database through the network pharmacology method, and found that because the two compounds have very similar structures, they have more common targets. According to the ranking of the total correlation scores, two chromone compounds were found to be more correlated with the inflammatory and apoptosis-related proteins ABCB1, ALOX5, NF-κB, Bcl-2 and Caspase9 (Table2). As we all know, a large number of studies have confirmed that the activation and response of NF-κB and Caspase pathways play an important role in the occurrence and development of gastric ulcer. Therefore, we selected the above targets to investigate the mechanism of action. WB results also showed that the two chromone compounds could significantly reduce the expression of inflammation and apoptosis-related proteins NF-κB, Caspase3 and Caspase9 (Figure 6). Immunohistochemical results also showed that the two chromone compounds could significantly reduce the expression of inflammatory and apoptotic proteins (Figure 5). Finally, the molecular docking results also showed that the functional groups of the two chromone compounds could highly bind to the amino acid residues of the related proteins (Figure 7). ABCB1 and ALOX5 are activated as upstream signaling molecules of NF-κB, which can activate the NF-κB pathway to induce inflammation. Caspase3, Caspase9 and Bcl-2 are key proteins in apoptosis pathway, and the two chromone compounds can regulate the expression of key proteins respectively, thereby inhibiting the inflammation and apoptosis of gastric ulcer tissue. We hope the present explanation can well explain the protective effect of chromone on gastric ulcer by affecting the expression of related pathways, thank you!

  1. In Figure 4, the effects of ulceration and treatment on serum cytokine levels are extremely modest. Please provide a rationale, and use an alternative inflammatory marker (e.g., serum CRP levels, or cytokines within the gastric tissue) to confirm these findings.

Response: Thanks for the comment of expert. A large number of studies have confirmed that tumor necrosis factor and interleukin-like cytokines are classical and commonly used indicators in the occurrence and development of inflammation, and the results show that the serum levels of these indicators are significantly increased in mice with gastric ulcer, and the tested drug agarwood chromone compounds can significantly reduce their expression, which can confirm the inhibition of inflammation caused by gastric ulcer., which can confirm the inhibitory effect on the inflammation caused by gastric ulcer. The CRP mentioned by experts is a non-specific inflammatory marker, which means that when the body tissue is damaged or the body is infected by a virus, it can reflect the problem of inflammation and infection of the body, and is generally a common indicator for the clinical diagnosis of chronic inflammatory diseases. We selected ethanol-induced acute gastric ulcer models, and CRP may not be a representative indicator. Part of the stomach tissue was photographed for the occurrence of gastric ulcer, part was observed by pathological section and immunohistochemistry, and part was detected by protein. So, there's no tissue use to test for inflammatory cytokines. The level of tissue inflammatory factors can more directly reflect the inflammation of gastric ulcer, and we will pay attention to this in the follow-up study, thank you!

  1. In Figure 6a, the western blots for 3 different proteins are showing the same bands of beta actin as control. This is unacceptable, and each protein must be normalized to its corresponding loading control (i.e., beta actin) on the same western blot membrane. For a phosphorylated protein, the corresponding whole protein can be used for normalization, instead of a loading control.

Response: Thanks for the comment of expert. When detecting protein expression through WB, we used the protein extracted from the same sample, and we also provided the original data of protein bands, which were detected three times respectively. Figure 5 shows the results of selected protein bands. For the phosphorylated protein of NF-κB, we purchased the phosphorylated protein, and the corresponding whole protein did not change significantly, so we used the internal reference protein as a loading control, I hope our explanation can be recognized by experts, thank you!

  1. Finally, the quality of English used needs to be greatly improved. As such, this manuscript would benefit from the services of a professional English language editor.

Response: Thanks for the comment of expert. We have revised and improved the grammar of our manuscript in detail, please refer to the revised manuscript for details. Thank you!

The above is our reply to all your comments, hope to be able to explain your questions well, and hope that our reply you get your approval. If you have any questions, please feel free to contact us. Thank you!

Best wishes’

Canhong Wang

Ling Zhang

Jianhe Wei

Reviewer 2 Report

Comments and Suggestions for Authors

The manuscript pharmaceuticals-3355689 investigated the network pharmacology and molecular docking of agarwood chromone anti-gastric ulcer via inhibiting the NF-κB and Caspase pathways.

Agarwood chromone as anti-gastric ulcer via inhibiting the NF-κB and Caspase pathways has already been investigated and research results have been published in scientific journals:

Alamil JMR, Paudel KR, Chan Y, Xenaki D, Panneerselvam J, Singh SK, Gulati M, Jha NK, Kumar D, Prasher P, Gupta G, Malik R, Oliver BG, Hansbro PM, Dua K, Chellappan DK. Rediscovering the Therapeutic Potential of Agarwood in the Management of Chronic Inflammatory Diseases. Molecules. 2022 May 9;27(9):3038. doi: 10.3390/molecules27093038. PMID: 35566388; PMCID: PMC9104417.

Wang C, Peng D, Liu Y, Wu Y, Guo P, Wei J. Agarwood Alcohol Extract Protects against Gastric Ulcer by Inhibiting Oxidation and Inflammation. Evid Based Complement Alternat Med. 2021 Sep 18;2021:9944685. doi: 10.1155/2021/9944685. PMID: 34580595; PMCID: PMC8464430.

Hang Zhang, Jia-Le Ma, Chuang Chang, He Ta, Yun-Fang Zhao, She-Po Shi, Yue-Lin Song, Peng-Fei Tu, Hui-Xia Huo, Jiao Zheng, Jun Li,

Gastroprotective 2-(2-phenylethyl)chromone-sesquiterpene hybrids from the resinous wood of Aquilaria sinensis (Lour.) Gilg, Bioorganic Chemistry, Volume 133, 2023, 106396,ISSN 0045-2068, https://doi.org/10.1016/j.bioorg.2023.106396.

Additional comments:

-In general, figures must be improved. Titles and scales are too small.

-Figure 6 needs to be clarified. Figure 6 (a) Target protein and reference protein banding map is a low-quality image. The low quality of WB analysis may be due to technical problems in critical steps. Quality control procedures should ensure robust data generation. Please revise:

Bass JJ, Wilkinson DJ, Rankin D, Phillips BE, Szewczyk NJ, Smith K, Atherton PJ. An overview of technical considerations for Western blotting applications to physiological research. Scand J Med Sci Sports. 2017 Jan;27(1):4-25. doi: 10.1111/sms.12702. Epub 2016 Jun 5. PMID: 27263489; PMCID: PMC5138151.

-Conclusions must be improved. Consider the most crucial idea you want your readers to take away with them after reading your paper. Include outlining further areas of inquiry.

Author Response

Review comments and responses

Dear reviewer,

Thank you very much for your review of our manuscript, recognition and valuable revision suggestions. We have made careful modifications according to your comments and explained them one by one, as follows:

Review comments 2:

Agarwood chromone as anti-gastric ulcer via inhibiting the NF-κB and Caspase pathways has already been investigated and research results have been published in scientific journals:

Alamil JMR, Paudel KR, Chan Y, Xenaki D, Panneerselvam J, Singh SK, Gulati M, Jha NK, Kumar D, Prasher P, Gupta G, Malik R, Oliver BG, Hansbro PM, Dua K, Chellappan DK. Rediscovering the Therapeutic Potential of Agarwood in the Management of Chronic Inflammatory Diseases. Molecules. 2022 May 9;27(9):3038. doi: 10.3390/molecules27093038. PMID: 35566388; PMCID: PMC9104417.

Wang C, Peng D, Liu Y, Wu Y, Guo P, Wei J. Agarwood Alcohol Extract Protects against Gastric Ulcer by Inhibiting Oxidation and Inflammation. Evid Based Complement Alternat Med. 2021 Sep 18; 2021:9944685. doi: 10.1155/2021/9944685. PMID: 34580595; PMCID: PMC8464430.

Hang Zhang, Jia-Le Ma, Chuang Chang, He Ta, Yun-Fang Zhao, She-Po Shi, Yue-Lin Song, Peng-Fei Tu, Hui-Xia Huo, Jiao Zheng, Jun Li. Gastroprotective 2-(2-phenylethyl)chromone-sesquiterpene hybrids from the resinous wood of Aquilaria sinensis (Lour.) Gilg, Bioorganic Chemistry, Volume 133, 2023, 106396, ISSN 0045-2068, https://doi.org/10.1016/j.bioorg.2023.106396.

Response: Thanks for the comment of expert. About the existing research on the anti-inflammation and anti-gastric ulcer of agarwood, we make the following explanations. The published articles listed above (including the study of our research group) are studies on the protective effects of agarwood components on chronic inflammation, gastric ulcers and gastric cell damage, which are different from our study. The tested drugs they used were not agarwood chromone compounds, but other components of agarwood, and the models of gastric ulcers were different and the pathways of mechanism of action were also different. In this study, based on previous studies, we further carried out research on the anti-gastric ulcer effect and mechanism of agarwood chromone compounds. In addition, we adopted modern advanced network pharmacology and molecular docking technology to confirm and analyze the anti-gastric ulcer effect and mechanism of the main pharmacodynamic chromone compounds of agarwood. To further reveal the anti-gastric ulcer mechanism of agarwood components and to provide a strong basis for the prevention and treatment of gastric ulcer diseases, thank you!

Additional comments:

-In general, figures must be improved. Titles and scales are too small.

Response: Thanks for the comment of expert. We provide original uncombined pictures for small or fuzzy pictures, which can be used by magazine typesetting and editing, and can display our research results more clearly and intuitively. Thank you!

-Figure 6 needs to be clarified. Figure 6 (a) Target protein and reference protein banding map is a low-quality image. The low quality of WB analysis may be due to technical problems in critical steps. Quality control procedures should ensure robust data generation. Please revise:

Bass JJ, Wilkinson DJ, Rankin D, Phillips BE, Szewczyk NJ, Smith K, Atherton PJ. An overview of technical considerations for Western blotting applications to physiological research. Scand J Med Sci Sports. 2017 Jan;27(1):4-25. doi: 10.1111/sms.12702. Epub 2016 Jun 5. PMID: 27263489; PMCID: PMC5138151.

Response: Thanks for the comment of expert. In view of the poor quality of protein bands, we did the following analysis and treatment. When detecting protein through WB, we strictly followed the procedure and repeated the experiment three times. We also provided the original protein strip when submitting. The reason for the unclear display may be the lower resolution of the strip processing. We can provide a clearer picture after modification, thank you!

-Conclusions must be improved. Consider the most crucial idea you want your readers to take away with them after reading your paper. Include outlining further areas of inquiry.

Response: Thanks for the comment of expert. According to the requirements of experts, we have made a more in-depth overview and summary of the conclusion, and further elaborated the more important information and the next research to be carried out, hoping to bring readers more useful and important information. See the revised draft for details. Thank you!

The above is our reply to all your comments, hope to be able to explain your questions well, and hope that our reply you get your approval. If you have any questions, please feel free to contact us. Thank you!

Best wishes’

Canhong Wang

Ling Zhang

Jianhe Wei

Reviewer 3 Report

Comments and Suggestions for Authors

accept

Reviewer 4 Report

Comments and Suggestions for Authors

The manuscript entitled “Agarwood chromone alleviate gastric ulcer via inhibiting the NF-κB and Caspase pathways based on Network pharmacology and molecular docking” has many mistakes, authors need to rectify many portions.

1.      What is the reason for providing Figure 9 after the conclusion?

2.      Remove the redundancy to improve readability and clarity. Why is the phrase "a famous" repeated in "a famous famous traditional Chinese medicine (TCM)"?

3.      Revise to "has a better curative effect in treating gastrointestinal diseases" for grammatical correctness.

4.      Use "chromone is a predominant constituent" or "chromone was a predominant constituent," depending on the temporal context.

5.      Rewrite to "However, the effect and mechanism of agarwood in treating gastrointestinal diseases remain unclear."

6.      Simplify to "Numerous studies, including ours, have found."

7.      Could "as shown in Figure 2a-B" be more specific about what Figure 2a-B illustrates?

8.      Specify the aspect in which CHR2 is superior, e.g., "CHR2 showed a higher ulcer inhibition rate compared to CHR1."

9.      Ensure consistency in terminology and grammar throughout the document.

10.   Define all abbreviations at their first occurrence.

11.   The author required to update recent references can be seen PMID: 37065061, 39065802, 35424125, 34297427, 35014595, 36936534.

12.   The methodology references are missing

13.   Why did the author convert sdf to mol and then pdbqt, while directly mol can be converted to pdbqt?

14.   What is the grid size? Did the author use the Autodock tool?

15.   Specify the pdbid for the use in the method and which residues were targeted for the docking study.

16.   The docking figure must be revised due to low resolution.

Overall, the Authors did nice work but did not explain nicely.

Good Luck!

Comments on the Quality of English Language

English language must be correct, few comments provided on it. 

Round 2

Reviewer 1 Report

Comments and Suggestions for Authors

Most of the concerns raised by this reviewer have been successfully addressed by the authors. However, a few issues remain:

1. The sources and clones of the various primary antibodies used in western blotting have been stated; but it is unclear if the revised text refers to primary antibodies. We suggest specifically mentioning "primary antibodies against..." before stating the names of antibodies as to avoid any confusion.

2. The name and source of secondary antibody used in western blotting must be provided.

3. In case of western blotting, each band must be presented above its corresponding loading control (beta actin). This has not yet been done in Figure 6 as requested (the authors rationale for not doing so is hard to understand). 

4. Finally, it is unclear why ROS was shown in the diagram presented in the end, as the manuscript did not examine any effects of ROS on ulceration or the actions of chromones. Please mention the significance of ROS in this context, or remove it entirely.

5. Finally, the authors need to mention potential reasons for the extremely modest changes observed in serum cytokine levels. Would it be possible to detect such cytokines through immunofluorescence (or immunohistochemistry) done on gastric tissue sections ?

Reviewer 2 Report

Comments and Suggestions for Authors

Figure 6 must be improved. The experiment needs to fulfill the quality standards to be accepted for publication. If it is impossible to improve, the paper can not be accepted.